# Urinary Oxidative Stress Biomarkers in the Diagnosis of Detrusor Overactivity in Female Patients with Stress Urinary Incontinence

**DOI:** 10.3390/biomedicines11020357

**Published:** 2023-01-26

**Authors:** Wei-Hsin Chen, Yuan-Hong Jiang, Hann-Chorng Kuo

**Affiliations:** 1Department of Urology, Hualien Tzu Chi Hospital, Buddhist Tzu Chi Medical Foundation, Hualien 970, Taiwan; 2Department of Urology, School of Medicine, Tzu Chi University, Hualien 970, Taiwan

**Keywords:** biomarkers, urodynamics, female, urinary bladder

## Abstract

Ninety-three women with urodynamic stress incontinence (USI) and a mean age of 60.8 ± 10.7 (36–83) years were retrospectively enrolled. According to their VUDS, 31 (33%) were grouped into USI and detrusor overactivity (DO), 28 (30.1%) were grouped into USI and hypersensitive bladder (HSB), and 34 (36.6%) were controls (USI and stable bladder). The USI and DO group had significantly increased 8-isoprostane (mean, 33.3 vs. 10.8 pg/mL) and 8-hydroxy-2-deoxyguanosine (8-OHdG; mean, 28.9 vs. 17.4 ng/mL) and decreased interleukin (IL)-2 (mean, 0.433 vs. 0.638 pg/mL), vascular endothelial growth factor (mean, 5.51 vs. 8.99 pg/mL), and nerve growth factor (mean, 0.175 vs. 0.235 pg/mL) levels compared to controls. Oxidative stress biomarkers were moderately diagnostic of DO from controls, especially 8-isoprostane (area under the curve (AUC) > 0.7). Voided volume was highly diagnostic of DO from either controls or non-DO patients (AUC 0.750 and 0.915, respectively). The proposed prediction model with voided volume, 8-OHdG, and 8-isoprostane (cutoff values 384 mL, 35 ng/mL, and 37 pg/mL, respectively) had an accuracy of 81.7% (sensitivity, 67.7%; specificity, 88.7%; positive predictive value, 75.0%; negative predictive value, 84.6%). Combined with voided volume, urinary oxidative stress biomarkers have the potential to be used to identify urodynamic DO in patients with USI.

## 1. Introduction

Detrusor overactivity (DO) is common in women with stress urinary incontinence (SUI) [1]. DO is a term of urodynamic study (UDS) and is well correlated with the symptom of urgency. However, an unneglectable portion of asymptomatic patients with coexisting DO and urodynamic urinary incontinence (USI) before anti-incontinence surgery has been reported [1]. Although DO does not preclude anti-incontinence surgery and patients with DO and SUI still have improved quality of life after a sling operation, the lesser degree of symptom improvement or even the risk of persistent urgency after the surgery still bothers patients with coexisting DO and mixed urinary incontinence [2,3]. Hence, identifying DO has an essential role in the preoperative evaluation [4].

The role of UDS, including video urodynamic study (VUDS), has been controversial over the past decade. The necessity of UDS before anti-incontinence surgery has been questioned, as only a few urodynamic parameters predict the treatment outcome of anti-incontinence surgery [5,6]. A randomized trial in which 630 patients with urinary incontinence were randomly assigned to the office evaluation only and UDS groups showed no significance in treatment satisfaction at the 1 year follow-up [7]. Yet, coexisting overactive bladder (OAB) and SUI may be a result of urethral instability-induced DO, in which urethral instability influences storage and voiding reflexes, as well as leads to urinary urgency [8]. In the diagnosis of urethral instability-induced DO (urethral-related OAB), VUDS is the only test to identify changes of urethral incompetence during bladder filling on abdominal straining or pressure increase [9]. The urine leaked into the proximal urethra during abdominal straining might induce reflexic detrusor contractions which mimic genuine SUI in patients with an incompetent urethra. VUDS before anti-incontinence surgery is justified and plays a role in diagnosing complicated SUI and preoperative planning [9].

However, VUDS is an invasive and less available diagnostic study in many countries and rural areas. Noninvasive alternative methods to help identify DO in women with SUI have not been explored. There is an unmet need for a noninvasive method to identify DO in women with SUI in clinical practice. Recently, several urinary biomarkers have been identified to be correlated with OAB [10]. With urinary biomarkers, DO can be suspected and identified in clinics. In this study, we aimed to evaluate whether urinary biomarkers can be used to identify DO in women with SUI.

## 2. Materials and Methods

### 2.1. Study Design

Ninety-three women with SUI who were diagnosed to have USI under VUDS from August 2017 to December 2021 were retrospectively enrolled. Patients with evidence of urinary tract infection under urinalysis were excluded. Patients with any history of neurological disorders or injuries such as myasthenia sclerosis, myelomeningocele, and spinal cord injuries were also excluded. Patients with pelvic organ prolapse, which results in bladder outlet obstruction under VUDS, were also excluded. All women were enrolled at outpatient clinic after VUDS diagnosis, and their presenting symptoms were collected during history taking. Urgency was defined as “the complaint of a sudden, compelling desire to pass urine which is difficult to defer” by International Incontinence Society (ICS) standardization and terminology committees [11]. After VUDS, the patients were categorized into three subgroups according to their bladder condition and urodynamic finding of USI: USI and DO, USI and hypersensitive bladder (HSB), and USI and stable bladder (control) groups. DO was diagnosed when involuntary detrusor contraction during the filling phase was noted under VUDS according to ICS standardization and terminology [11]. HSB was diagnosed when the cystometric bladder capacity was less than 350 mL without evidence of uninhibited contraction of the detrusor muscle during bladder filling. USI was defined as the presence of urine leakage with or without bladder base hypermobility (descent of bladder neck more than 2 cm under VUDS) during the cough test. All VUDS diagnoses were made by a single urologist (HCK) according to ICS recommendations [12].

This study was approved by the Institutional Review Board and Ethics Committee of Buddhist Tzu Chi General Hospital (IRB-107-175-A, IRB-105-156-A, and IRB-105-31A). Each patient was informed about the study rationale and procedures, and written informed consent was obtained.

### 2.2. Urinary Biomarker Analysis

Urine samples were collected before VUDS and anti-incontinence surgery. Urine samples were self-voided after full bladder sensation was achieved. Cases of urinary tract infection were excluded on the basis of the results of the urinalysis (presence of white blood cell count >10/HPF). The methodology of the urine sample preparation, storage, and analyses was described elsewhere [13].

### 2.3. Quantification of 8-OHdG, 8-Isoprostane, and TAC

The targeted urinary oxidative stress biomarkers included total antioxidant capacity (TAC), 8-isoprostane, and 8-hydroxy-2-deoxyguanosine (8-OHdG). The quantification of 8-OHdG, 8-isoprostane, and TAC in the urine samples was performed in accordance with the respective manufacturer’s instructions (8-OHdG: ELISA kit, BioVision, Boston, MA, USA; 8-isoprostane: ELIZA kit, Enzo, NY, USA; TAC Assay Kit, Abcam, Boston, MA, USA). The laboratory processes were similar to those previously reported [14].

### 2.4. Quantification of Inflammatory Cytokines

The targeted inflammatory cytokines included interleukin-1β (IL-1β), interleukin-2 (IL-2), interleukin-6 (IL-6), interleukin-8 (IL-8), tumor necrosis factor-alpha (TNF-α), vascular endothelial growth factor (VEGF), brain-derived neurotrophic factor (BDNF), prostaglandin E 2 (PGE2), and neural growth factor (NGF). To measure urine inflammatory cytokines, we used the MILLIPLEX^®^ human cytokines/chemokines, adipokine, and neurodegenerative disease magnetic bead panels. The laboratory processes used to quantify these targeted analytes were described elsewhere [10].

### 2.5. Statistical Analysis

Continuous data were presented as the mean ± standard deviations, whereas categorical data were represented as numbers and percentages. For each targeted analyte, outliers were excluded from further analyses and defined as values outside the range between means ± 3 standard deviations in the study or control group. The demographical continuous data such as age, BMI, and UDS parameters between the study and control groups were compared by analysis of variance. The analytes between the three groups were analyzed using the Kruskal–Wallis test, and the post hoc paired comparison was performed with Bonferroni–Dunn correction.

The cutoff value and area under the curve (AUC) of the investigated urinary biomarkers were calculated by receiver operating characteristic (ROC) curve analysis. Sensitivity and specificity were determined according to Youden’s index. All of the abovementioned calculations were run in IBM^®^ SPSS^®^ version 25. A *p*-value < 0.05 was considered significant. Multivariate logistic regression models were created to adjust for confounding factors for each targeted analyte and to discover the optimal AUC under different combinations of urinary biomarkers. Linear regression analysis with Pearson correlation was performed to determine the correlation between urodynamic parameters and urine levels of the targeted biomarkers.

We established a decision tree model via RStudio, version 2022.7.2 + 576 (RStudio Team (2022), Boston, MA, USA) for post hoc analysis. To build the decision tree model, the data were stratified and partitioned into training data and test data with a ratio of 85% and 15%, respectively. Training data were used in modeling and test data were used in internal validation. A classification tree was carried out with train data via rpart package, and visualization of the tree was conducted with rpart.plot package. Pruning of the decision tree was performed with optimal complexity parameter (cp value) to avoid overfitting. Internal validation of the decision tree was performed with the test data.

## 3. Results

Ninety-three patients with a mean age of 60.8 ± 10.7 (range 36–83) years were analyzed. Thirty-one (33.3%) patients were included in the USI and DO group, 28 (30.1%) were included in the USI and HSB group, and 34 (36.6%) were included in the control group. Table 1 shows the clinical characteristics and UDS parameters of the three groups. There were no significant differences in age and body mass index (BMI) among these three groups. A significantly higher proportion of USI and DO group patients had a previous history of anti-incontinence surgery than the other two groups (*p* = 0.005). Regarding the underlying diseases, the proportion of patients with diabetes mellitus (DM) as comorbidity and with a history of cardiovascular accident (CVA) was the highest in the USI and DO groups (*p* = 0.018 and *p* = 0.012). Regarding the symptoms before anti-incontinence surgery that were relevant to DO, only the proportion of patients with urgency was significantly different among the three groups (*p* = 0.02). The proportion of patients with urgency was the highest (80.6%) in the USI and DO group, followed by that of USI and HSB patients at 63.0%, with that of controls (47.1%) being the lowest. For the UDS parameters, the USI and DO group had significantly increased detrusor pressure at maximal urinary flow (Pdet) (*p* = 0.008) compared with the control group. They also had significantly decreased maximum flow rate (Qmax) (*p* = 0.001), voided volume (vol) (*p* < 0.001), first sensation of filling (FSF) (*p* = 0.007), full sensation (FS) (*p* < 0.001), cystometric bladder capacity (CBC) (*p* < 0.001), and voided efficiency (VE) (*p* = 0.046) than controls. Among the aforementioned UDS parameters, Qmax and voided volume were the only two significant parameters that could be obtained in the office setting (pressure-flow study). The USI and HSB group also had significantly decreased Qmax, voided volume, FS, bladder compliance, and CBC than the controls.

Analytes with significantly different urine levels among the USI and DO, USI and HSB, and control groups included IL-2, VEGF, NGF, 8-isoprostane, and 8-OHdG (Table 2 and Figure 1). Compared with controls, the USI and DO group had significantly increased urine 8-isoprostane (33.3 ± 34.6 vs. 10.8 ± 10.4 pg/mL, *p* = 0.014) and urine 8-OHdG (28.9 ± 17.3 vs. 17.4 ± 11.7 ng/mL, *p* = 0.021), but decreased urine IL-2 (0.433 ± 0.212 vs. 0.638 ± 0.272 pg/mL, *p* = 0.021), urine VEGF (5.51 ± 4.99 vs. 8.99 ± 5.83 pg/mL, *p* = 0.015), and urine NGF (0.175 ± 0.083 vs. 0.235 ± 0.103 pg/mL, *p* = 0.027) levels. The distribution of urine IL-1β levels was different among these three groups (USI and DO 0.722 ± 0.229 vs. USI and HSB 0.717 ± 0.287 vs. controls 0.599 ± 0.235 pg/mL, *p* = 0.043). However, no significance was reported in the post hoc comparison between groups. Only the urine VEGF level in USI and HSB group was significantly distinct from that of the controls (5.68 ± 5.42 vs. 8.99 ± 5.83 pg/mL, *p* = 0.015). No significant differences in the levels of urinary analytes were found between the USI and DO and USI and HSB groups in the post hoc analysis.

Table 3 shows the diagnostic values of urinary biomarkers and voided volume, comparing the USI and DO group with the control group, including AUC using ROC analysis, cutoff values, sensitivity, specificity, positive predictive value (PPV), and negative predictive value (NPV) using Youden’s index. Urine 8-isoprostane was the only analyte with an AUC >0.7 with unimpressive sensitivity (67.7%) and specificity (70.6%). The AUCs of urine IL-1β, IL-2, VEGF, and 8-OHdG were all ≤0.7, with urine VEGF showing the highest specificity (90.9%). Voided volume had an AUC >0.9 with high sensitivity (96.8%) and specificity (87.9%).

Table 4 shows the diagnostic values of urinary biomarkers and voided volume, comparing the USI and DO group with the USI and non-DO group (USI and HSB and control groups). No urinary analytes had an AUC > 0.7. The AUCs of urine IL-1β, IL-2, VEGF, NGF, 8-isoprostane, and 8-OHdG were all <0.7 but >0.6. Among the abovementioned urinary analytes, 8-OHhdG had the highest specificity (88.7%). The AUC of voided volume was 0.75 with high sensitivity (96.8%) and NPV (97.1%).

Table 5 shows the results of the multivariate analysis of the targeted urinary biomarkers adjusted for age, comorbidity of DM, and history of CVA and anti-incontinence surgery using binary logistic regression. Urine 8-isoprostane and 8-OHdG were the only two urinary biomarkers showing significance between USI and DO and controls. No urinary biomarker was found to be a significant factor that distinguishes the USI and DO group from the USI and non-DO group in the multivariable analysis. Voided volume had a significant but weak correlation with certain urinary biomarkers, including L-1β, IL-2, TNF-α, NGF, 8-isoprostane, and 8-OHdG, in patients with USI and DO (Table 6).

We built a decision tree model to better diagnose DO in women with SUI. Voided volume and urinary biomarkers that were significantly distinct between groups in Table 2 (L-1β, IL-2, VEGF, NGF, 8-isoprostane, and 8-OHdG) were utilized to construct the decision tree. Before pruning, the accuracy tested with test data was 90% and the number of splits was four. The visualization of the decision tree is shown in Figure 2. In the decision tree, to distinguish DO patients from non-DO patients in the SUI women, voided volume was identified as the root node with a cutoff value of 384 mL, while the levels of urinary 8-OHdG and 8-isoprostane were identified as inner nodes with cutoff values of 35 ng/mL and 37 pg/mL, respectively. The overall accuracy of the decision tree was 80.7% in model training and 81.7% in internal validation. Diagnostic values including sensitivity, specificity, PPV, and NPV generated from model training were 64.3%, 89.1%, 75%, and 83.1%, respectively, while those from internal validation were 67.7%, 88.7%, 75%, and 84.6%, respectively.

## 4. Discussion

To our knowledge, this is one of the pioneering studies to explore the diagnostic values of urinary biomarkers for distinguishing DO from non-DO in women with SUI. In this study, the prediction of urodynamic DO in patients with SUI and urgency symptoms was possible using voided volume <384 mL, urine levels of 8-OHdG ≥35 ng/mL, and urine 8-isoprostane level ≥37 pg/mL with an accuracy of 81.7% (sensitivity 67.7%, specificity 88.7%, PPV 75%, and NPV 84.6%). With relatively high specificity and NPV, this result provides evidence that urinary oxidative stress biomarkers could be a useful tool to diagnose urodynamic DO in SUI women with SUI.

Women with SUI usually have frequency and urgency symptoms in addition to urinary incontinence while straining. The cause of the urgency frequency symptoms in women with SUI might result from urine leak into the proximal urethra, increased bladder sensation, and spontaneous DO during bladder filling or reaching bladder capacity. DO is a diagnosis by VUDS that features involuntary provoked or spontaneous detrusor contractions during the filling phase, while an overactive bladder is a clinical diagnosis that features urinary urgency with or without frequency, nocturia, and incontinence. Previous studies showed that the rate of DO in patients with OAB was 64% [15]. The pathophysiology of OAB could be neurogenic, inflammation, bladder outlet obstruction, or idiopathic [16]. Diabetes mellitus is an independent risk factor for OAB, and cardiovascular processes are prevalent in OAB patients [16]. Overexpression of sensory receptors and C-afferents, and increased excretion of sensory neuropeptides have been reported [16,17,18,19]. An imbalance of oxidative stress due to bladder ischemic and reperfusion injury, bladder outlet obstruction, and other factors has also been addressed [18]. Urinary biomarkers of increased oxidative stress, such as 8-OHdG and F2-isoprostane, were found to be elevated in human and animal models [18].

HSB is a urodynamic condition characterized by an early desire to void without fear of leakage or pain. Yamaguchi et al. suggested OAB as a hypersensitivity disorder given the fact that 43% of OAB patients in their cohort reported urgency less than once per day, and that bladder hypersensitivity was observed regardless of urgency episodes [20]. Although HSB patients seldom have urgency urinary incontinence, they still have mild urgency, which is inhibited by guarding. HSB patients might have bladder sensory hyperactivity, which might, in some patients, have a direct or indirect connection with increased bladder inflammation and sensory nerve activation [19,21]. Therefore, HSB patients might also have an increase in oxidative stress as a result of bladder inflammation or ischemia, which is not solely related to USI [20]. Although there were no significant differences in the level of the abovementioned biomarkers between the USI and HSB group and control groups, the pattern of elevated 8-OHdG and 8-isoprostane, increased IL-1β, and decreased IL-2 in patients with USI and HSB resembles that of patients with USI and DO, suggesting that patients with USI and HSB also have a bladder environment of increased inflammation and oxidative stress. Further studies are needed to explore the differences and correlations between DO and HSB in terms of pathophysiology and treatment outcomes.

8-Isoprostane is among the most thoroughly studied F2-isoprostanes, which are the end-products of lipid peroxidation stimulated by reactive oxygen species [22]. 8-Isoprostane is also among the most commonly targeted urinary biomarkers for the evaluation of oxidative stress related to detrusor contraction in humans and murine in vivo [23]. 8-OHdG is a stable end-product of DNA oxidation, the levels of which are not influenced by long-term storage of the urine specimen, even at −20 °C [24]. Women with OAB have significantly increased urine 8-OhdG [25]. In this study, patients with coexisting USI and DO had elevated urine 8-isoprostane and 8-OHdG levels, indicating that a higher bladder oxidative stress existed in patients with USI and DO, which was associated with a higher Pdet in this group of patients, suggesting that active detrusor contractility might exist.

Patients with DO and HSB had an increased bladder sensation and reduced bladder capacity [26]. In the bladder with urodynamic DO, bladder hypersensitivity also coexists [27]. Of all the significantly different urodynamic parameters, reduced voided volume was the only parameter that could be measured in clinics via uroflowmetry. This is the first study to utilize machine learning models such as a decision tree to build a diagnostic algorithm for DO in women with SUI. Several analyses including logistic regression can also increase diagnostic values by combining multiple variables, but the decision tree was among those that offer cutoff values. According to the decision tree established on the basis of this voided volume and urinary biomarker, SUI patients with voided volume <384 mL, urine 8-OHdG level ≥35 ng/mL, and urine 8-isoprostane level ≥37 pg/mL are more likely to have urodynamic DO. The accuracy of the established decision tree model for diagnosing DO in SUI patients was 81.7% (sensitivity 67.7%, specificity 88.7%, PPV 75%, and NPV 84.6%). With the ever-increasing availability and affordability of immunoassay for urinary biomarkers, diagnosing DO in patients with SUI in clinics has the potential to be more prevalent.

This study had several limitations. Our sample size was small. This was a single-centered retrospective study, which was not immune to selection and observational biases. The result of the study could also be confounded with the disproportional distribution of comorbidities such as diabetes mellitus among different groups. In addition, the urine levels of inflammatory cytokines tended to be less stable than oxidative stress biomarkers, including 8-isoprostane, 8-OHdG, and TAC. Some urine samples were stored for a certain period of time, and urine protein degradation might have occurred over time. External validation of the proposed decision tree is needed in the future.

## 5. Conclusions

Urine 8-isoprostane and 8-OHdG levels were higher in patients with USI and DO than in patients with USI and a stable bladder. Patients with USI and DO also had a significantly reduced voided volume. With a combination of voided volume, urine 8-isoprostane, and 8-OHdG, we might be able to identify occult DO in patients with SUI.

## Figures and Tables

**Figure 1 biomedicines-11-00357-f001:**
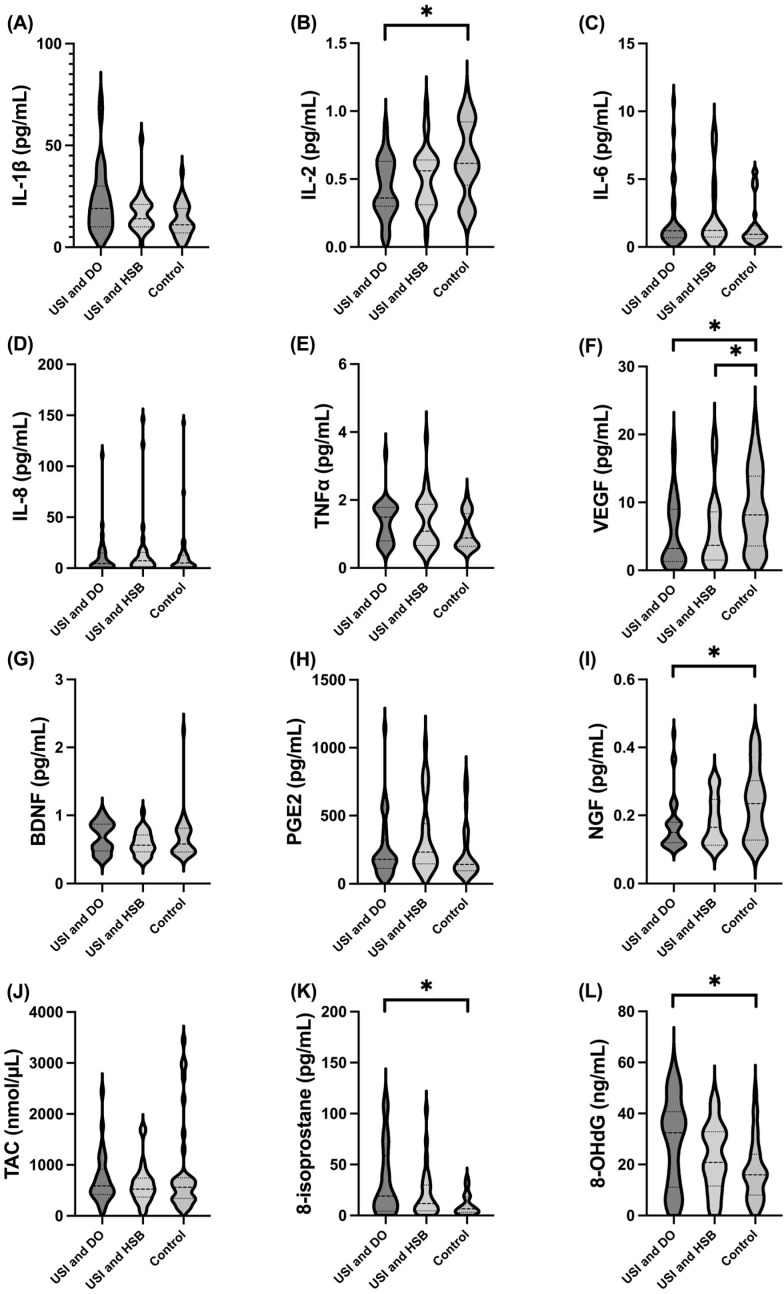
Violin plots of the levels of urinary biomarkers in the USI and DO, USI and HSB, and control groups (USI and stable bladder). Urine levels of (**B**) IL-2, (**F**) VEGF, (**I**) NGF, (**K**) 8-isoprostane, and (**L**) 8-OHdG were significantly different in paired comparison among the USI and DO, and control groups. Urine levels of (**F**) VEGF were significantly different among the USI and HSB and control groups. DO and USI, detrusor overactivity; USI, urodynamic stress incontinence; HSB, hypersensitive bladder; (**A**) IL-1β, interleukin-1β; (**B**) IL-2, interleukin-2; (**C**) IL-6, interleukin-6; (**D**) IL-8, interleukin-8; (**E**) TNF-β, tumor necrosis factor-beta; (**F**) VEGF, vascular endothelial growth factor; (**G**) BDNF, brainderived neurotrophic factor; (**H**) PGE2, prostaglandin E2; (**I**) NGF, neural growth factor; (**J**) TAC, total antioxidant capacity; (**K**) 8-isoprostane; (**L**) 8-OHdG, 8-hydroxy-2-deoxyguanosine. * *p* < 0.05.

**Figure 2 biomedicines-11-00357-f002:**
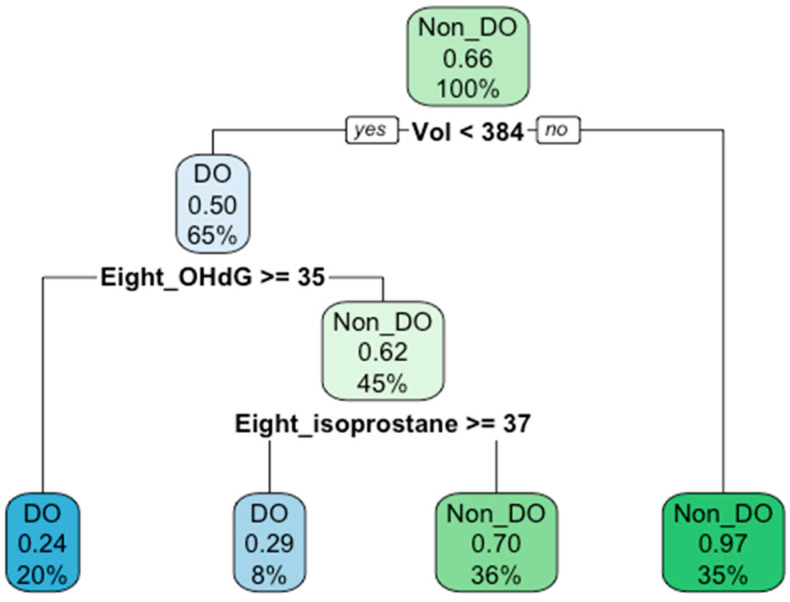
The decision tree model for distinguishing DO patients from non-DO patients with USI. The root node comprises voided volume (Vol) and the inner nodes include 8-OHdG (presented as Eight_OHdG) and 8-isoprostane (presented as Eight_isoprostane). Decimal numbers in the middle of each node represent the ratio of non-DO data split by the node in the training model. Percentage numbers at the bottom of each node represent the percentage for which the data in the node account in the training model. The accuracy of the training model and internal validation was 80.72% and 81.72%, respectively. DO: detrusor overactivity; Non_DO, Non-DO; Vol, voided volume; Eight_OHdG, 8-hydroxy-2-deoxyguanosine; Eight_isoprostane, 8-isoprostane.

**Table 1 biomedicines-11-00357-t001:** Clinical characteristics and urodynamic parameters of the USI and DO, USI and HSB, and control groups (USI and stable bladder).

Parameter	(A) USI + DO	(B) USI + HSB	(C) Control	Total	*p*-Value	**Post Hoc**
	(*N* = 31)	(*N* = 28)	(*N* = 34)			
**Age (years)**	61.9 ± 11.4	59.6 ± 10.5	60.6 ± 10.5	60.8 ± 10.7	0.714	
**BMI**	27.5 ± 5.12	25.9 ± 4.52	24.6 ± 4.71	26.0 ± 4.90	0.06	
**Previous SUI surgery**	6 (19.40%)	1 (3.60%)	0 (0.00%)	7 (7.50%)	0.005	
**HTN**	17 (54.8%)	9 (32.1%)	12 (35.3%)	38 (40.9%)	0.148	
**DM**	11 (35.5%)	4 (14.3%)	3 (8.8%)	18 (19.4%)	0.018	
**CVA**	5 (16.1%)	4 (14.3%)	0 (0%)	9 (9.7%)	0.012	
**Cancer**	3 (9.7%)	2 (7.1%)	1 (2.9%)	6 (6.5%)	0.535	
**Autoimmune diseases**	1 (3.2%)	0 (0%)	2 (5.9%)	3 (3.2%)	0.293	
**Urgency**	25 (80.6%)	17 (63.0%)	16 (47.1%)	58 (62.4%)	0.020	
**Pdet (cmH_2_O)**	23.0 ± 16.3	16.6 ± 9.9	13.3 ± 8.3	17.7 ± 12.7	0.008	A vs. C
**Qmax (mL/s)**	14.5 ± 7.5	13.5 ± 6.2	20.2 ± 7.3	16.2 ± 7.6	0.001	A vs.C, B vs. C
**Volume (mL)**	262 ± 85.0	267 ± 115	446 ± 122	330 ± 139	<0.001	A vs. C, B vs. C
**PVR (mL)**	36.7 ± 61.1	38.5 ± 86.5	6.9 ± 14.2	27.2 ± 62.0	0.094	
**FSF (mL)**	121 ± 37.1	145 ± 54.2	166 ± 69.1	145 ± 58.1	0.007	A vs. C
**FS (mL)**	200 ± 65.0	236 ± 79.9	298 ± 97.7	247 ± 91.7	<0.001	A vs. C, B vs. C
**Compliance**	73.5 ± 61.7	95.2 ± 55.1	178 ± 111	118 ± 93.1	<0.001	A vs. C, B vs. C
**BCI**	95.2 ± 36.6	82.7 ± 34.6	109.9 ± 40.5	96.8 ± 38.8	0.02	B vs. C
**CBC (mL)**	298 ± 90.0	304 ± 93.1	439 ± 141.2	351 ± 130	<0.001	A vs. C, B vs. C
**cQmax**	0.86 ± 0.42	0.76 ± 0.32	0.95 ± 0.31	0.86 ± 0.36	0.118	
**VE**	0.90 ± 0.17	0.87 ± 0.28	0.98 ± 0.04	0.92 ± 0.19	0.046	A vs. C

USI: urodynamic stress incontinence, DO: detrusor overactivity, HSB: hypersensitive bladder; HTN, hypertension; DM, diabetes mellitus; CVA, cardiovascular accident; Pdet, detrusor pressure at maximal flow; Qmax, maximum flow rate; PVR, post-void residual volume; FSF, first sensation of filling; FS, first sensation; BCI, bladder contractility index; CBC, cystometric bladder capacity; cQmax, contracted maximal flow; VE, voiding efficiency (voided volume divided by bladder capacity).

**Table 2 biomedicines-11-00357-t002:** Levels of urinary biomarkers among the USI and DO, USI and HSB, and control groups.

Urinary Biomarkers ^@^	(A) USI + DO	(B) USI + HSB	(C) Control	Total	*p*-Value	**Post Hoc**
	(*N* = 31)	(*N* = 28)	(*N* = 34)			
**IL-1** **β**	0.722 ± 0.229 (1)	0.717 ± 0.287 (0)	0.599 ± 0.235 (2)	0.677 ± 0.255 (3)	0.043	
**IL-2**	0.433 ± 0.212 (0)	0.516 ± 0.222 (1)	0.638 ± 0.272 (0)	0.533 ± 0.252 (1)	**0.021**	A vs. C
**IL-6**	2.15 ± 2.61 (0)	2.41 ± 2.60 (0)	1.62 ± 1.67 (0)	2.03 ± 2.30 (0)	0.312	
**IL-8**	12.4 ± 21.3 (1)	17.8 ± 34.2 (0)	13.3 ± 26.8 (1)	14.4 ± 27.5 (2)	0.867	
**TNF** **α**	1.37 ± 0.621 (1)	1.37 ± 0.795 (0)	1.06 ± 0.509 (1)	1.26 ± 0.655 (2)	0.128	
**VEGF**	5.51 ± 4.99 (0)	5.68 ± 5.42 (0)	8.99 ± 5.834 (0)	6.81 ± 5.62 (0)	**0.015**	A vs. C, B vs. C
**BDNF**	0.678 ± 0.220 (1)	0.588 ± 0.177 (0)	0.674 ± 0.334 (1)	0.649 ± 0.257 (2)	0.272	
**TAC**	725 ± 496 (1)	611 ± 388 (1)	878 ± 911 (0)	748 ± 663 (2)	0.791	
**PGE2**	256 ± 236 (1)	335 ± 270 (0)	221 ± 200 (0)	267± 237 (1)	0.063	
**NGF**	0.175 ± 0.083 (0)	0.183 ± 0.072 (1)	0.235 ± 0.103 (0)	0.199 ± 0.091 (1)	**0.027**	A vs. C
**8-Isoprostane**	33.3 ± 34.6 (2)	21.5 ± 24.5 (0)	10.8 ± 10.5 (0)	21.5 ± 26.4 (2)	**0.014**	A vs. C
**8-OHdG**	28.9 ± 17.3 (0)	22.4 ± 12.9 (0)	17.4 ± 11.7 (0)	22.6 ± 14.7 (0)	**0.021**	A vs. C

USI: urodynamic stress incontinence, DO: detrusor overactivity, HSB: hypersensitive bladder; IL-1β, interleukin-1β; IL-2, interleukin-2; IL-6, interleukin-6; IL-8, interleukin-8; TNF-α, tumor necrosis factor-alpha; VEGF, vascular endothelial growth factor; BDNF, brainderived neurotrophic factor; TAC, total antioxidant capacity; PGE2, prostaglandin E2; NGF, neural growth factor; 8-OHdG, 8-hydroxy-2-deoxyguanosine. Values in parentheses indicate the number of outliers. ^@^ Units: all pg/mL, except nmol/μL in TAC, and ng/mL in 8-OHdG.

**Table 3 biomedicines-11-00357-t003:** Diagnostic values of urinary biomarkers and voided volume between the USI and DO and control groups.

Urinary Biomarkers ^@^	AUC	Cutoff Value ^@^	Sensitivity (%)	Specificity (%)	PPV (%)	NPV (%)
**IL-1** **β**	0.680	>0.665	63.33%	68.75%	66%	67%
**IL-2**	0.698	<0.545	67.74%	67.65%	66%	70%
**IL-6**	0.555	>1.175	51.72%	66.67%	58%	61%
**IL-8**	0.506	<1.795	33.33%	78.79%	59%	57%
**TNF** **α**	0.644	>0.940	70%	57.58%	60%	78%
**VEGF**	0.690	<2.090	41.94%	90.91%	81%	63%
**BDNF**	0.558	>0.515	73.33%	45.45%	55%	65%
**TAC**	0.528	>354.480	86.67%	38.24%	55%	76%
**PGE2**	0.680	<0.190	77.42%	58.82%	63%	74%
**NGF**	0.569	>157.555	66.67%	64.71%	63%	69%
**8-Isoprostane**	0.705	>12.005	67.74%	70.59%	68%	71%
**8-OHdG**	0.684	>24.855	62.07%	82.35%	75%	72%
**Voided volume**	0.915	<384.0	96.77	87.88	88%	97%

AUC, area under curve; PPV, positive predictive value; NPV, negative predictive value; IL-1β, interleukin-1β; IL-2, interleukin-2; IL-6, interleukin-6; IL-8, interleukin-8; TNF-α, tumor necrosis factoralpha; VEGF, vascular endothelial growth factor; BDNF, brainderived neurotrophic factor; TAC, total antioxidant capacity; PGE2, prostaglandin E2; NGF, neural growth factor; 8-OHdG, 8-hydroxy-2-deoxyguanosine. ^@^ Units: all pg/mL, except nmol/μL in TAC, ng/mL in 8-OHdG, and mL in volume.

**Table 4 biomedicines-11-00357-t004:** Diagnostic values of urinary biomarkers and voided volume between the USI and DO and non-DO groups (USI and HSB and control groups).

Urinary Biomarkers ^@^	AUC	Cutoff Value ^@^	Sensitivity (%)	Specificity (%)	PPV (%)	NPV (%)
**IL-1** **β**	0.606	>0.695	56.7%	66.7%	45.9%	75.5%
**IL-2**	0.655	<0.515	61.3%	65.6%	47.5%	76.9%
**IL-6**	0.506	>1.180	51.7%	58.3%	37.5%	71.4%
**IL-8**	0.519	<1.785	33.3%	77.1%	41.7%	70.1%
**TNF** **α**	0.578	>0.940	70.0%	54.1%	42.9%	78.6%
**VEGF**	0.604	<2.090	41.9%	80.3%	52.0%	73.1%
**BDNF**	0.588	>0.745	50.0%	72.1%	46.9%	74.6%
**TAC**	0.539	>365.889	86.7%	31.2%	38.2%	82.6%
**PGE2**	0.507	<71.440	20.0%	93.6%	60.0%	70.7%
**NGF**	0.618	<0.190	77.40%	50%	43.6%	81.6%
**8-Isoprostane**	0.654	>12.495	62.3%	67.7%	48.0%	79.0%
**8-OHdG**	0.657	>33.955	48.3%	88.7%	66.7%	78.6%
**Voided volume**	0.750	<384.0	96.8%	54.1%	51.7%	97.1%

AUC, area under curve; PPV, positive predictive value; NPV, negative predictive value; IL-1β, interleukin-1β; IL-2, interleukin-2; IL-6, interleukin-6; IL-8, interleukin-8; TNF-α, tumor necrosis factoralpha; VEGF, vascular endothelial growth factor; BDNF, brain-derived neurotrophic factor; TAC, total antioxidant capacity; PGE2, prostaglandin E2; NGF, neural growth factor; 8-OHdG, 8-hydroxy-2-deoxyguanosine; Vol, voided volume. ^@^ Units: all pg/mL, except nmol/μL in TAC, ng/mL in 8-OHdG, and mL in volume.

**Table 5 biomedicines-11-00357-t005:** Multivariate analysis of the targeted urinary biomarkers (adjusting for age, DM, and history of CVA and anti-incontinence surgery).

Urinary Biomarkers ^@^	*p*-Value	Odds Ratio	95% CI
**USI + DO vs. control**			
**IL-1** **β**	0.607	2.656	0.064–110.6
**IL-2**	0.093	0.024	0.000–1.851
**VEGF**	0.452	0.934	0.780–1.117
**NGF**	0.230	3386.128	0.006–1000
**8-Isoprostane**	0.038	1.046	1.002–1.090
**8-OHdG**	0.019	1.064	1.010–1.120
**USI + DO vs. USI + non-DO**			
**IL-1** **β**	0.700	1.566	0.159–15.40
**IL-2**	0.183	0.111	0.004–2.819
**VEGF**	0.585	1.035	0.916–1.169
**NGF**	0.920	0.623	0.000–1000
**8-Isoprostane**	0.262	1.013	0.990–1.037
**8-OHdG**	0.097	1.032	0.994–1.071

IL-1β, interleukin-1β; IL-2, interleukin-2; VEGF, vascular endothelial growth factor; NGF, neural growth factor; 8-OHdG, 8-hydroxy-2-deoxyguanosine. Non-DO includes HSB and controls. ^@^ Units: all pg/mL, except ng/mL in 8-OHdG.

**Table 6 biomedicines-11-00357-t006:** The correlation coefficient (r value) between the urinary biomarker levels and clinical characteristics of patients with USI and DO.

Urinary Biomarkers ^@^	Pdet	Qmax	Vol	PVR	FSF	FS	Compliance	BCI	CBC	cQmax	VE
**IL-1** **β**	n.s.	n.s.	−0.218	n.s.	n.s.	n.s.	n.s.	n.s.	n.s.	n.s.	n.s.
**IL-2**	−0.220	0.223	0.377	n.s.	n.s.	n.s.	n.s.	n.s.	0.329	n.s.	n.s.
**IL-6**	n.s.	n.s.	n.s.	n.s.	n.s.	−0.271	n.s.	n.s.	n.s.	n.s.	n.s.
**IL-8**	n.s.	n.s.	n.s.	n.s.	n.s.	n.s.	n.s.	n.s.	n.s.	n.s.	n.s.
**TNF** **α**	0.212	−0.261	−0.337	0.416	n.s.	n.s.	n.s.	n.s.	n.s.	−0.213	−0.391
**VEGF**	n.s.	n.s.	n.s.	n.s.	n.s.	n.s.	n.s.	n.s.	0.208	n.s.	n.s.
**BDNF**	n.s.	n.s.	n.s.	n.s.	n.s.	n.s.	n.s.	n.s.	n.s.	n.s.	n.s.
**TAC**	n.s.	n.s.	n.s.	n.s.	n.s.	n.s.	n.s.	n.s.	n.s.	n.s.	n.s.
**PGE2**	n.s.	n.s.	n.s.	n.s.	n.s.	n.s.	n.s.	n.s.	n.s.	n.s.	n.s.
**NGF**	−0.247	0.212	0.335	n.s.	n.s.	0.216	n.s.	n.s.	0.259	n.s.	n.s.
**8-Isoprostane**	n.s.	n.s.	−0.391	n.s.	n.s.	n.s.	n.s.	n.s.	−0.333	n.s.	−0.256
**8-OHdG**	n.s.	n.s.	−0.249	n.s.	n.s.	n.s.	n.s.	n.s.	n.s.	n.s.	n.s.

IL-1β, interleukin-1β; IL-2, interleukin-2; IL-6, interleukin-6; IL-8, interleukin-8; TNF-α, tumor necrosis factor-alpha; VEGF, vascular endothelial growth factor; BDNF, brain-derived neurotrophic factor; TAC, total antioxidant capacity; PGE2, prostaglandin E2; NGF, neural growth factor; 8-OHdG, 8-hydroxy-2-deoxyguanosine; Pdet, detrusor pressure at maximal flow; Qmax, maximum flow rate; Vol, voided volume; PVR, post-void residual volume; FSF, first sensation of filling; FS, first sensation; BCI, bladder contractility index; CBC, cystometric bladder capacity; cQmax, contracted maximal flow; VE, voiding efficiency (voided volume divided by bladder capacity); n.s., not significant. ^@^ Units: all pg/mL, except nmol/μL in TAC, and ng/mL in 8-OhdG.

## Data Availability

The data that support the findings of this study are available on request from the corresponding author.

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
