# Peer review of "Urinary Oxidative Stress Biomarkers in the Diagnosis of Detrusor Overactivity in Female Patients with Stress Urinary Incontinence"

_biomedicines, 2023, doi:10.3390/biomedicines11020357_

Round 1
Reviewer 1 Report
The authors evaluated urinary oxidative stress biomarkers in female patients with stress urinary incontinence and assessed their potential role in the diagnosis of mixed detrusor overactivity and intrinsic sphincter deficiency. The involvement of oxidative stress in this disorder is too briefly overviewed in the article in order to clarify the implication of this pathogenic mechanism. What are the mechanisms linking oxidative stress and stress urinary incontinence? Were there any correlations between the oxidative stress markers assessed and the levels of pro-inflammatory cytokines? What other comorbidities did these subjects suffer from? Oxidative stress and inflammation are involved in a myriad of diseases, e.g., diabetes, obesity, metabolic syndrome, hypertension, rheumatological disorders, cardiovascular disease, psoriasis, cancer etc. See:10.3390/antiox11020282 In diabetes, for example, there is an increase in reactive oxygen species and a decrease in antioxidant levels. You need to evaluate the impact of comorbidities on your findings, probably by regressions/multivariable analyses.
Author Response
Please refer to the affiliated file (Response to reviewer1) and the revised manuscript.

Reviewer 2 Report
Thank you for this important and novel study. Here are a few suggestions
1. If you could produce a nomogram that greatly advance the application to general practice.
2.. If you could further clarify how your measures are similar or different from ICS measures, that would be an improvement.
3. I understand that there is no page limit, so adding sentences for clarification would help in many places.
Abst
First sentence is about DO and SUI. Second sentence brings in 3 new ideas; confusing, please revise, 1 sentence for each new idea.
35 persistent urgency to persistent urgency after surgery
42 yet, coexisting - - please clarify at many places if you are talking about before or after surgery
45 In patients with urethral stability-induced DO – this is a new term and needs definition including urodynamic classification.
47. Thus, VUDS still has a role in diagnosing complicated SUI and preoperative planning of anti-incontinence surgery in patients with urethral-related OAB or urethral instability. BETTER to say because of evidence,1,2,3, for surgery, VUDS justified in certain cases.
51 what is “eyeball urodynamic study”
52. what are - - Other non-invasive alternative methods to help identify DO in clinics are still unexplored.
55. With the availability of commercial kits for urinary biomarkers, what is evidence, needs more support.
64 Patients with pelvic organ prolapse, which results in bladder outlet obstruction, were also excluded. CAN you clarify this factor? It presents with a wide range that can confound all results, discussion.
67. All symptom data were collected at the clinic. NOTE 3 day voiding diary is standard of care; please justify why it was not used.
68. The patients were categorized into three groups according to their detrusor condition: ISD and DO, ISD and hypersensitive bladder (HSB), and ISD and stable bladder (control) groups. PLEASE compare with (ICS) standardization and terminology.
75. ISD was defined as the presence of urine leakage without evident bladder base hypermobility during the cough test during VUDS. Please define hypermobility (+/-).
85. Infections were excluded based on the results of the urinalysis before urine samples were stored. Please clarify UTI assessment.
106. For each targeted analyte, outliers were 105 excluded from further analyses. Do you report all cases of exclusion in your results?
120 partitioned into train data and test data with a ratio of 85% and 15% respectively. Please clarify train and test data.
128. Thirty-one (33.3%), 28 (30.1%), and 34 (36.6%) patients were included in the ISD 128 and DO, ISD and HSB, and control groups, respectively. Problem with 3 and 4 groups in same sentence, please clarify.
149. Analytes with significantly different urine levels among the ISD and DO, ISD and 149 HSB, and control group included IL-1β, IL-2, VEGF, NGF, 8-isoprostane, and 8-OHdG 150 (Table 2 and Figure 1). PROBLEM, looking at fig 1, the differences for the 3 groups are not apparent, please explain – add arrows tp highlight differences - mean values would help.
Author Response
Please refer to the affiliated file (Response to reviewer2) and the revised manuscript.

Round 2
Reviewer 1 Report
The authors have accurately revised the paper.